# Finite Element Analysis Could Predict and Prevent a Pathological Femoral Shaft Fracture after En Bloc Resection of a Large Osteoid Osteoma

**DOI:** 10.3390/children9020158

**Published:** 2022-01-26

**Authors:** Tadashi Iwai, Naoto Oebisu, Manabu Hoshi, Naoki Takada, Hiroaki Nakamura

**Affiliations:** Department of Orthopedic Surgery, Osaka City University Graduate School of Medicine, 1-4-3 Asahi-Machi, Abeno-Ku, Osaka 545-8585, Japan; evis@med.osaka-cu.ac.jp (N.O.); hoshi@med.osaka-cu.ac.jp (M.H.); m2026957@med.osaka-cu.ac.jp (N.T.); hnakamura@med.osaka-cu.ac.jp (H.N.)

**Keywords:** osteoid osteoma, en bloc resection, artificial bone graft, O-arm, fracture, finite element analysis

## Abstract

Osteoid osteoma is a benign tumor. Approximately 20% of these tumors are located in the femur. The tumor primarily occurs in children and is treated by surgical excision or radiofrequency ablation. Recently, bone-tumor resection using three-dimensional (3D) intraoperative imaging with an O-arm in combination with a navigation system has been reported to be effective. However, there is a risk of postoperative fracture because of the weakening of the bone after drilling for tumor resection. A 12-year-old Japanese girl presented with an osteoid osteoma in the left femoral shaft, which resulted in a fracture after en bloc resection and artificial bone grafting using a 3D image-guided (O-arm) assisted navigation system. Orthopedic oncologists should be aware of the risk of fracture. Moreover, they should consider the mechanical risk prediction of bone fracture using finite element analysis prior to treatment.

## 1. Introduction

An osteoid osteoma is the third most frequent benign bone tumor, constituting 10–12% of primary bone tumors, and usually affects children and adolescents [1]. The tumor appears as a small lesion (<2 cm) consisting of a radiolucent nidus of immature osteoid surrounded by reactive sclerotic bone [1]. Approximately 20% of osteoid osteomas are located in the proximal femur [2]. A combination of clinical features and typical image findings on plain radiographs, computed tomography (CT), and magnetic resonance imaging (MRI) is adequate for a reliable diagnosis in most cases [3]. Concerning the imaging method, it has also been widely used for the diagnosis of various illnesses [4,5,6]. Typically, patients with osteoid osteoma experience severe pain at night, which often responds effectively to nonsteroidal anti-inflammatory drugs (NSAIDs) [7]. A surgical excision may be required for the management of pain refractory to NSAIDs [8]. While the success rate of surgical excision is high (almost 100%) [9], the recent introduction of radiofrequency ablation (RFA) or microwave ablation has provided a less invasive treatment method with a relatively high success rate [10,11]. However, RFA or microwave ablation can damage the neighboring organs and is contraindicated for cases of osteoid osteomas adjacent to crucial neurovascular tissues [12]. Furthermore, RFA is not approved by the National Health Insurance in Japan. Therefore, the medical care cost is high in cases where RFA is performed. Regardless of the treatment type, an accurate localization of the nidus is essential during the procedure [12]. Over the years, several percutaneous techniques using CT guidance have been used to reduce the surgical morbidity associated with open procedures [8]. A recent report has indicated the advantage of using three-dimensional (3D) intraoperative imaging with an O-arm in combination with a CT-based navigation system [13]. Using this technique, they could accurately resect bone tumors in an anatomical position without radiation exposure [13]. Major complications of the procedure are rare. However, there is a risk of postoperative fracture because of the weakening of the bone after drilling for tumor resection. This is the first case report showing that the treatment of an osteoid osteoma using an O-arm-assisted navigation system resulted in a postoperative fracture.

## 2. Case Presentation

A 12-year-old Japanese girl was admitted to the hospital complaining of sharp pain at night, localized in the left femur. In addition, the patient had no medical history. Regarding the activities of daily living, there were also no issues noted. Radiographic and CT findings revealed a 10-mm × 4.5-mm × 3-mm lesion located in the lateral aspect of the left femoral shaft. The lesion had a radiolucent nidus surrounded by a sclerotic bone (Figure 1). 

Moreover, a low-signal lesion was visible on both the T1-weighted and T2-weighted sequence MRI images of the left femoral shaft. A high density polygonal-shaped lesion was observed in the center of the lesion on the T2-weighted sequence using MRI (Figure 2). 

The patient was administered NSAIDs for pain relief; however, the pain would still wake her up from sleep, thereby limiting daily activity. Although a biopsy had not been performed, these clinical and radiographic features were completely compatible with an osteoid osteoma. The treatment options were explained to the patient, and the patient was advised on O-arm-guided en bloc resection and artificial bone grafting. After consent was obtained from the patient and her family, the procedure was performed. General anesthesia was administered to induce unconsciousness during the procedure. Under O-arm navigation, a 2-mm tract was drilled through the nidus from the lateral to the medial side (Figure 3A). Subsequently, the lateral cortex was drilled completely, while the medial cortex was partially drilled to remove the tumor using a 12-mm cannulated cutter (approximately 15% of the circumference, presented in Figure 3B,C). Based on previous reports [14,15], after the completion of en bloc bone resection, including the bone tumor, an artificial bone graft (β-tricalcium phosphate block, 10 mm × 20 mm × 10 mm, superpore standard-type: Pentax New Ceramics Division, Hoya Corporation, Tokyo, Japan) was performed for the bone defect (Figure 3D,E).

Pathological examination findings confirmed the diagnosis of osteoid osteoma (Figure 4).

After surgery, the patient was advised to perform non-weight-bearing activities for 4 weeks. Subsequently, when she attempted partial weight-bearing, she noticed an immediate snap along with a stabbing pain. Pelvic and left femur radiography was immediately performed, which showed a femoral shaft fracture (Figure 5A).

The fracture occurred because of full weight-bearing on the femur while she was putting on her shoes. The subsequent radiographic images matched perfectly with the old ones regarding the location of the osteoid osteoma and the fracture level (Figure 5A). Based on previous reports that focused on the management of pediatric diaphysial femoral fractures [16,17], intramedullary k-wire fixation using 3.0 mm Kirschner wires was performed for the fracture due to the patient’s constrained financial situation (Figure 5B).

The patient and her family were informed that the data from this case would be submitted for publication, and they provided their consent.

## 3. Discussion

Recently, minimally invasive techniques using a 3D image-guided (O-arm) system have become the most widely used and validated treatment method. O-arm-guided percutaneous excision is an alternative surgical technique. O-arm navigation provides a 3D view that permits the surgeon to find the safest path to the lesion under navigation without radiation exposure. A needle is inserted into the nidus during O-arm navigation. A small incision is made, and a biopsy punch is inserted through the needle, followed by the removal of the specimen. 

According to a previous systematic review of cases that received minimally invasive treatment for osteoid osteoma of the long bone, postoperative fractures after CT-guided RFA or percutaneous bone resection and drilling under CT guidance have been reported [18]. Such fractures have been reported in patients who had undergone treatment for lesions in the weight-bearing bones [19]. It is essential to obtain knowledge on the material properties of the bone and biomechanics of the skeleton under a physiological load to evaluate the fracture risk related to this treatment. When the bone strength is decreased by the surgical procedure performed, the fracture may be caused by a similar compressive loading. This concept may be especially crucial in the femoral shaft due to its anterior bow and biomechanics.

In the present case, the O-arm-guided en bloc resection was selected as not only a safe but also an effective method for the surgery of osteoid osteomas; however, this minimally invasive procedure does not entirely eliminate the risk of fracture. The tumor location and size of the drilled hole were considered the factors that had mainly resulted in a postoperative fracture. In view of the material properties of the bone, biomechanics of the femur, and risk of stress risers, the lateral femoral shaft should be approached with caution when treating osteoid osteomas. The affected bone can respond to form a sclerotic region, which is harder, but not necessarily stronger. This region may be considered a locus of minoris resistentiae, especially in the lower limbs. Restrictions on physical activities after the treatment are also highly advised, although a definitive statement regarding the appropriate duration for such restrictions cannot be made. It is also recognized that compliance is difficult in a young and active patient population. Therefore, mechanical risk prediction of bone fractures using a finite element analysis (FEA) prior to treatment may be necessary.

With respect to the efficacy of FEA, it was possible to predict risk factors for pathological fractures after bone tumor biopsies [20]. In the present case, the drilled hole was 12 mm (approximately 15% of the circumference). Based on the previous report, the bone strength after the procedure would have been approximately half. After surgery, an FEA was retrospectively performed to examine the patient’s bone strength before and after the treatment (Figure 6A,B); based on the CT images, a 3D FE model of the femur was recreated by Mechanical Finder version 9.0 (Research Center of Computational Mechanics, Inc., Tokyo, Japan). The femoral trabecular bone and inner part of the cortex were meshed by linear tetrahedral elements with a global edge length of 1.5 mm. The outer surface of the cortical bone was modeled by three nodal-point shell elements with a 0.3-mm thickness. The CT value of each element was decided as the average of the voxels contained in one element. Additionally, mechanical properties of each element were calculated in Hounsfield units [21]. Previous reports had used Keller’s equation for FEA; therefore, this equation was also used [22,23].

The following specific equations were used:

Young’s modulus (E, MPa).
E = 0.001 (ρ = 0)
E = 1890 ρ1.92 (ρ < 0).

Yield stress (σ, MPa).
σ = 1.0 × 1020 (ρ ≤ 0.2).
σ = 284 ρ2.27 (ρ > 0.2).

Modulus values <0.01 MPa were defined as 0.01 MPa, while those >20 GPa were defined as 20 GPa [23]. Poisson’s coefficient for each element was set at 0.3 [23]. 

According to the results, the FE-predicted fracture loads were 4000 N and 1800 N, respectively. This result coincided with the previous report [20]. The present case demonstrates the efficacy of FEA while raising awareness regarding the risk of fractures of the femoral shaft. However, FEA should have been performed preoperatively to predict and prevent the catastrophic results experienced by this young patient.

The optimal sequence for this 12-year-old girl could be:Clinical and radiological evaluation of the osteoid osteoma.FEA to predict bone mechanical properties after the bone resection with the decided method, modified by tumor dimensions.Tumor resection and femoral stabilization, according to results of FEA.

## 4. Conclusions

Orthopedic oncologists should be aware of the risk of fractures even after performing a minimally invasive surgery, such as O-arm-guided percutaneous excision. Generally, it is necessary to verify the optimal size of the drilled hole in order to avoid causing postoperative fractures by conducting FEA before the surgical procedure for osteoid osteoma.

## Figures and Tables

**Figure 1 children-09-00158-f001:**
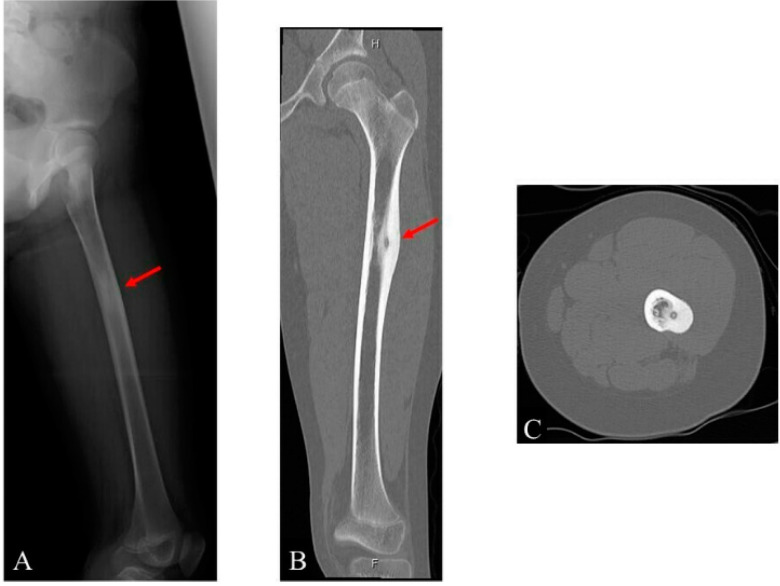
Images of the left femur of a 12-year-old girl: (**A**) radiograph; (**B**) coronal computed tomography (CT) image; and (**C**) axial CT image.

**Figure 2 children-09-00158-f002:**
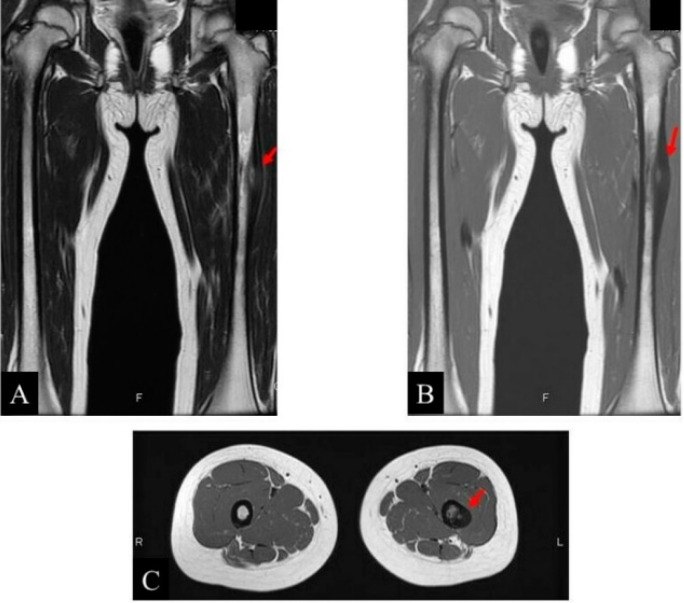
Images of the left femur of a 12-year-old girl: (**A**) coronal magnetic resonance imaging (MRI) T1-weighted image; (**B**) coronal MRI T2-weighted image; and (**C**) axial MRI T2-weighted image.

**Figure 3 children-09-00158-f003:**
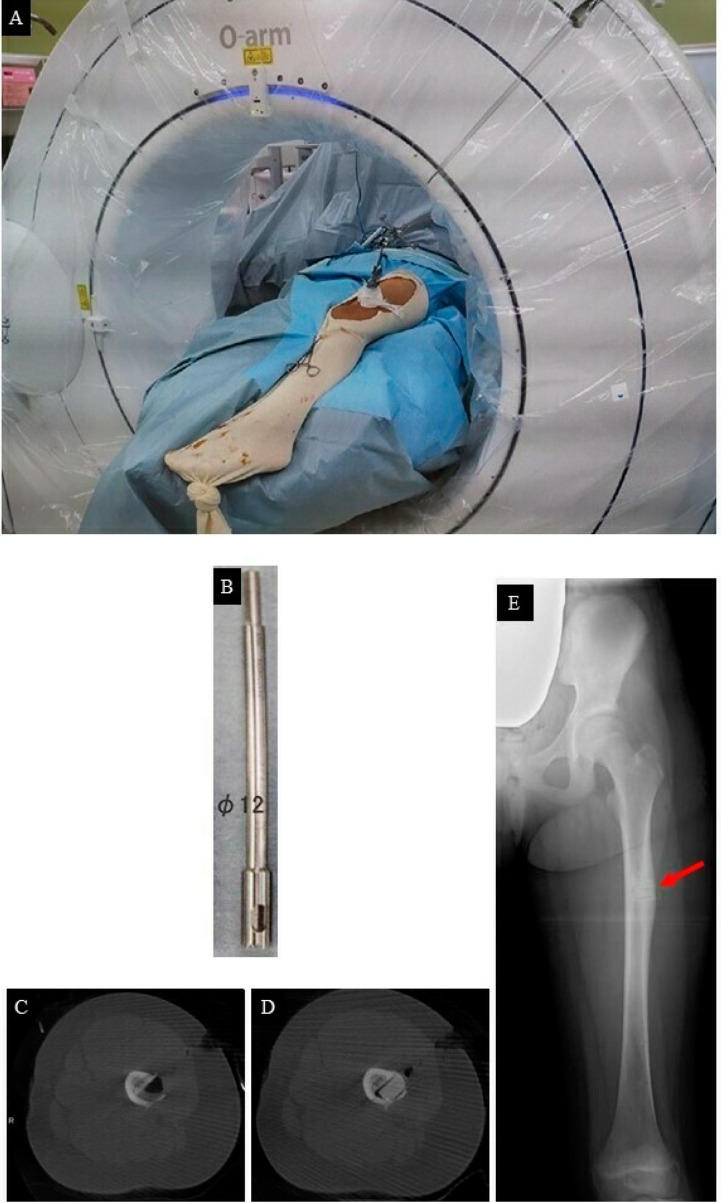
(**A**) O-arm-guided en bloc resection; (**B**) a 12-mm cannulated cutter; (**C**) the bone tumor was resected; (**D**) β-tricalcium phosphate block was grafted; and (**E**) postoperative radiograph.

**Figure 4 children-09-00158-f004:**
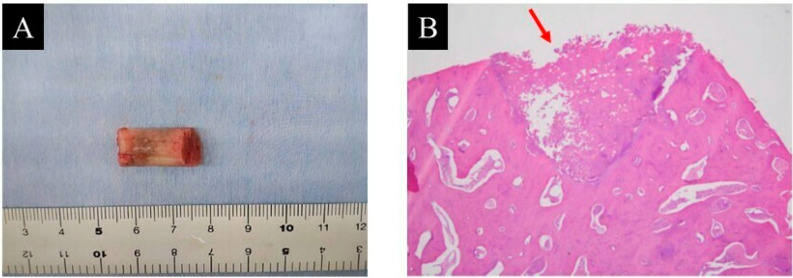
(**A**) resected specimen and (**B**) pathological examination (hematoxylin and eosin staining; magnification ×20).

**Figure 5 children-09-00158-f005:**
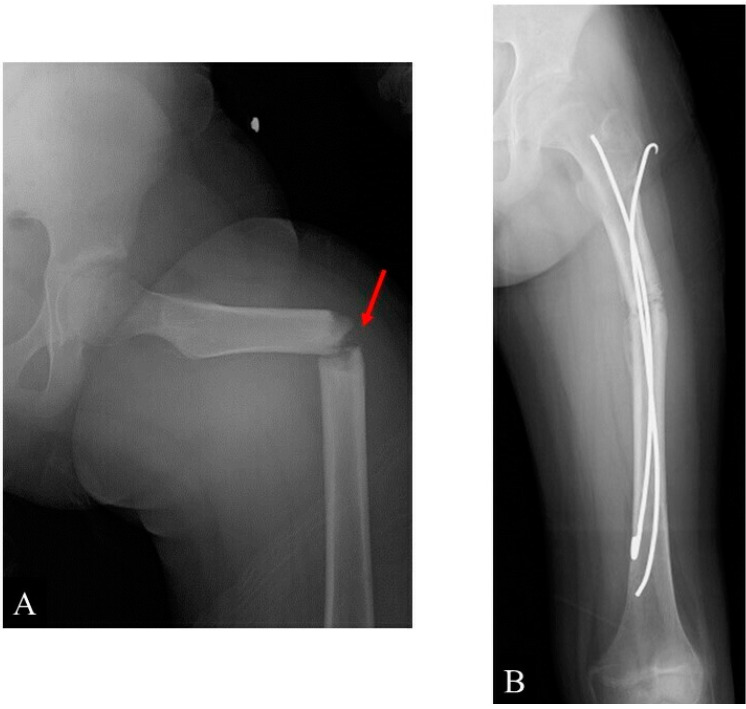
Radiographs: (**A**) a fracture after the O-arm-guided en bloc resection and artificial bone graft and (**B**) post-intramedullary k-wire fixation. Callus formation can be seen.

**Figure 6 children-09-00158-f006:**
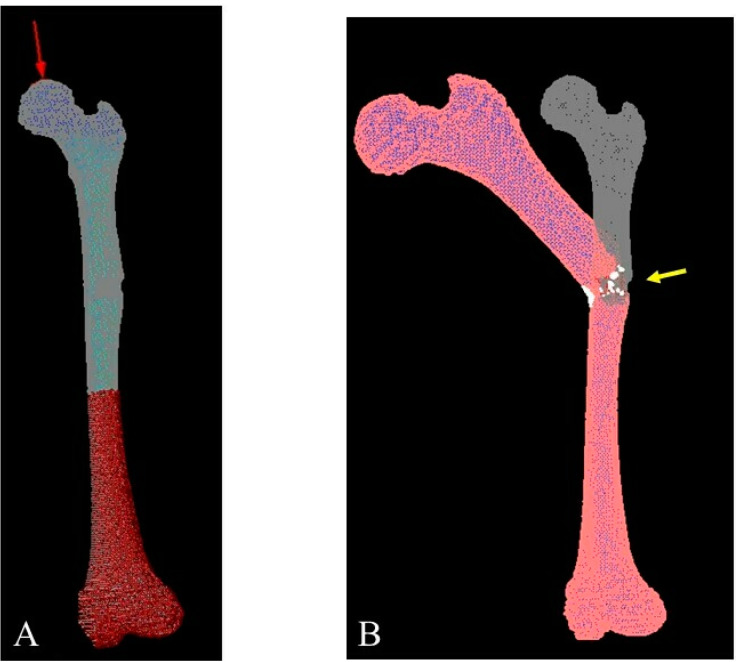
A three-dimensional finite element (FE) femur model was developed from computed tomography images. (**A**) The direction of compression was parallel to the mechanical axis. (**B**) FE analysis using the FE model was ended when fractured.

## Data Availability

All the data are available from the corresponding author upon reasonable request.

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
