# Peer review of "Finite Element Analysis Could Predict and Prevent a Pathological Femoral Shaft Fracture after En Bloc Resection of a Large Osteoid Osteoma"

_children, 2022, doi:10.3390/children9020158_

Round 1

Reviewer 1 Report

Dear author,
The article ’A Femoral Shaft Fracture after En Bloc Resection and Artificial Bone Graft for Osteoid Osteoma using an Intraoperative O-Arm-Assisted Navigation System’is well written, nevertheless i invite you to increase the paper. 
Preferably the length of a published Research articles should be of 4000-6000 words with 75 or more references excluding figures, structures, photographs, schemes, tables, etc.
 I suggest you to increase the Introduction focusing more on the imaging method (CT MRI etc). (line 30)

  • Mandibular coronoid process hypertrophy: Diagnosis and 20-year follow-up with CBCT, MRI and EMG evaluations DOI: 10.3390/app11104504
  • Unilateral superior condylar neck fracture with dislocation in a child treated with an acrylic splint in the upper arch for functional repositioning of the mandible   PMID: 27398739
  • Temporomandibular disc displacement with reduction treated with anterior repositioning splint: A 2-year clinical and magnetic resonance imaging (MRI) follow-up PMID: 32064850

You describe more in detail the pain therapy trough NSAIDs,describe the adverse reactions caused by these drugs used extensively by these child patients and the possible therapeutic alternatives with natural molecules that can reduce many side effects(line 33)

  • Polyphenols as potential agents in the management of temporomandibular disorders
    DOI: 10.3390/APP10155305

Finally the readers would be facilitated in reading by List of abbreviations.

Good luck and best Regards 

Author Response

Dear author,

The article ’A Femoral Shaft Fracture after En Bloc Resection and Artificial Bone Graft for Osteoid Osteoma using an Intraoperative O-Arm-Assisted Navigation System’is well written, nevertheless i invite you to increase the paper.

Preferably the length of a published Research articles should be of 4000-6000 words with 75 or more references excluding figures, structures, photographs, schemes, tables, etc.

Response: The authors would like to thank the reviewer for his/her constructive critique to improve the manuscript. We have made every effort to address the issues raised and to respond to all comments. The revisions are indicated in red font in the revised manuscript. Please, find next a detailed, point-by-point response to the reviewer's comments. We hope that our revisions would meet the reviewer’s expectations.

We agree with the reviewer that it is necessary to write a research article. We are now increasing the number of cases and preparing new articles. However, it will take a long time to establish the evidence. To our knowledge, this is the first article showing that the treatment of an osteoid osteoma using an O-arm-assisted navigation system resulted in a postoperative fracture. Therefore, the manuscript was submitted as a Case Report.

 I suggest you to increase the Introduction focusing more on the imaging method (CT MRI etc). (line 30)

Mandibular coronoid process hypertrophy: Diagnosis and 20-year follow-up with CBCT, MRI and EMG evaluations DOI: 10.3390/app11104504

Unilateral superior condylar neck fracture with dislocation in a child treated with an acrylic splint in the upper arch for functional repositioning of the mandible   PMID: 27398739

Temporomandibular disc displacement with reduction treated with anterior repositioning splint: A 2-year clinical and magnetic resonance imaging (MRI) follow-up PMID: 32064850

Response: We would like to thank the reviewer for the suggestion. Please note that we have revised the Introduction section and cited the suggested articles (References #4–6).

You describe more in detail the pain therapy trough NSAIDs,describe the adverse reactions caused by these drugs used extensively by these child patients and the possible therapeutic alternatives with natural molecules that can reduce many side effects(line 33)

Polyphenols as potential agents in the management of temporomandibular disorders

DOI: 10.3390/APP10155305

Response: We would like to thank the reviewer for the comment. Please note that we have corrected the Introduction section and cited the suggested article (Reference #8).

Finally the readers would be facilitated in reading by List of abbreviations.

Response: We would like to thank the reviewer for the insightful suggestion. As per the reviewer’s suggestion, we have added the list of abbreviations.

Good luck and best Regards

Response: We would like to thank the reviewer for evaluating our manuscript. We hope that our revisions would meet the reviewer’s expectations.

Reviewer 2 Report

Add a full stop before "for" in line 10 to improve sentence.
Correct ab-lation to ablation in line 11.
In lines 11-13 authors refer to previous report indicating the validity of image guided surgeries.
However in line 16- 17 the authors list the point as a part of their conclusion. Therefore I recommend removing the statement from the conclusion or at least re phrasing it as an emphasis on previous reports.
The authors bring up the point of finite element analysis which is in my opinion a very valid point however it was not very clearly formulated in the abstract, as the authors know finite element analysis requires a certain level off software understanding and cannot be apart of preoperative planning without considering time limitations especially in cases of trauma the reviwer suggests that the authors tackle this point in their discussion.
Be the animal parts line 49 authors are recommended to add to our knowledge this is the first case study to .......
In the case presentation then description of figure 3A in the text it's not quite accurate figure 3A does not show 12 millimeter cutter,  the images in figure 3 do not indicate the bone graft figure 3 legend is not quite representative to the images the review recommends that the authors improve the figure legends throughout the paper.
Also authors are encouraged to improves images nd figures through labelling the important parts they expect the reader to see.
Second the author should also provide a little bit more information about the Fe analysis methodology and the parameters that were used in it specially regarding the surface mesh.

Author Response

The authors would like to thank the reviewer for his/her constructive critique to improve the manuscript. We have made every effort to address the issues raised and to respond to all comments. The revisions are indicated in red font in the revised manuscript. Please, find next a detailed, point-by-point response to the reviewer's comments. We hope that our revisions would meet the reviewer’s expectations.

Add a full stop before "for" in line 10 to improve sentence.

Response: We would like to thank the reviewer for the comment. Please note that we have made this correction, as per the reviewer’s suggestion.

Correct ab-lation to ablation in line 11.

Response: We would like to thank the reviewer for the comment. Please note that we have made this correction, as per the reviewer’s suggestion.

In lines 11-13 authors refer to previous report indicating the validity of image guided surgeries.

However in line 16- 17 the authors list the point as a part of their conclusion. Therefore I recommend removing the statement from the conclusion or at least re phrasing it as an emphasis on previous reports.

Response: We would like to thank the reviewer for the comment. Please note that we have made this correction, as per the reviewer’s suggestion.

The authors bring up the point of finite element analysis which is in my opinion a very valid point however it was not very clearly formulated in the abstract, as the authors know finite element analysis requires a certain level off software understanding and cannot be apart of preoperative planning without considering time limitations especially in cases of trauma the reviwer suggests that the authors tackle this point in their discussion.

Response: We would like to thank the reviewer for the suggestion. Please note that we have added the following sentence to the Discussion section of the revised manuscript:

“based on the CT images, a 3D FE model of the femur was recreated by Mechanical Finder version 9.0 (Research Center of Computational Mechanics, Inc., Tokyo, Japan). The femoral trabecular bone and inner part of the cortex were meshed by linear tetrahedral elements with a global edge length of 1.5 mm. The outer surface of the cortical bone was modeled by three nodal-point shell elements with a 0.3-mm thickness. The CT value of each element was decided as the average of the voxels contained in one element. Mechanical properties of each element were calculated in Hounsfield units [20]. Previous reports had used Keller’s equation for FEA; therefore, this equation was also used [21, 22]. The following specific equations:

Young’s modulus (E, MPa).

E=0.001 (ρ=0)

E=1890 ρ1.92 (ρ <0).

Yield stress (σ, MPa).

σ=1.0 × 1020 (ρ ≤0.2).

σ=284 ρ2.27 (ρ >0.2).

Modulus values <0.01 MPa were defined as 0.01 MPa, while those >20 GPa were defined as 20 GPa [22]. Poisson’s coefficient for each element was set at 0.3 [22].” (Lines 137–154)

Be the animal parts line 49 authors are recommended to add to our knowledge this is the first case study to .......

In the case presentation then description of figure 3A in the text it's not quite accurate figure 3A does not show 12 millimeter cutter,  the images in figure 3 do not indicate the bone graft figure 3 legend is not quite representative to the images the review recommends that the authors improve the figure legends throughout the paper.

Also authors are encouraged to improves images nd figures through labelling the important parts they expect the reader to see.

Response: We would like to thank the reviewer for the suggestion. Please note that we have improved the quality of all figures.

Second the author should also provide a little bit more information about the Fe analysis methodology and the parameters that were used in it specially regarding the surface mesh.

Response: We would like to thank the reviewer for the suggestion. Please note that we have collected and added the information of FEA.

Reviewer 3 Report

Dear authors,

This topic could be interesting if it was clear. It is a case report targeting to an unknown and unclear target. Studying your paper I was not sure if you recommend or not intraoperative O-arm-Assisted Navigation system for osteoid osteoma resection. Some serious flaws were detected and some points need clarification.

Consent Form: This form is signed by the 12 year old patient and not by her parents, while in lines 64-65 is written that also her parents consented for the procedure and for publication (lines 96-97). Patient is underaged worldwide and cannot decide by herself legally.

Lines 28-29: Johan et al mentioned that "20% of osteoid osteomas are located in the proximal femur" and generally in the femur as is witten in the paper.

Line 53: Please provide data regarding the general health status of the patient. Moreover in the x-rays distal femoral epiphysis does not seem normal, especially regarding the medial femoral condyle. 

Lines 54-55: The size of the tumor as it is described is much larger than what has been already reported regarding this procedure (Fujiwara et al 2018). Is there any possibility of malpractice exceeding the limits of the procedure?

Lines 66-69: The technique has been performed in slightly different way than was described in the study of Fujiwara et al (2018) which was used as reference. In your case both cortices (lateral where tumor was located and medial) were drilled, affecting negatively the result.

Lines 70-71: Artificial bone graft was preferred over naive bone graft (allograft or autograft). Please explain which kind of artificial bone graft was used, why and the quantity of it as it cannot be located in the postoperative x-ray.

Lines 85-86: Two cortices were drilled and almost â…“ of the femoral diameter is missing while the rest of the femoral cortex has been split in the middle. This condition caused bone weakening. Please provide data supporting the decision for not internal or external fixation immediately after the tumor resection.  Moreover, is available an x-ray at 4-week postoperatively according to it patient was advised for partial weight-bearing? 

Lines 94-95: Please provide reference supporting the treatment of femoral fracture with k-wires. Even the use of titanium elastic nails is contraindicated for patients with unstable femoral fracture weighted more than 49 kg (Khoriati Al-achraf, Jones Carl, Gelfer Gael, Trompeter Alex The management of paediatric diaphysial femoral fractures: a modern approach. 

Strat Traum Limb Recon (2016) 11:87–97 DOI 10.1007/s11751-016-0258-2)

Lines 117-139: This part of discussion is irrelevant to the subject. Finite element analysis (FEA) could be useful preoperatively to prevent the fracture and the pain and the second surgery. The technology, the knowledge and the know-how was available to you as it is obvious from the self-citation 

(Iwai, T.; Hoshi, M.; Oebisu, N.; Orita, K.; Shimatani, A.; Takada, N.; Nakamura, H. Prediction of risk factors for pathological fracture after bone tumor biopsy using finite element analysis. Cancer Manag Res 2021, 13, 3849–3856)

Author Response

Dear authors,

This topic could be interesting if it was clear. It is a case report targeting to an unknown and unclear target. Studying your paper I was not sure if you recommend or not intraoperative O-arm-Assisted Navigation system for osteoid osteoma resection. Some serious flaws were detected and some points need clarification.

Response: The authors would like to thank the reviewer for his/her constructive critique to improve the manuscript. We have made every effort to address the issues raised and to respond to all comments. The revisions are indicated in red font in the revised manuscript. Please, find next a detailed, point-by-point response to the reviewer's comments. We hope that our revisions would meet the reviewer’s expectations.

Consent Form: This form is signed by the 12 year old patient and not by her parents, while in lines 64-65 is written that also her parents consented for the procedure and for publication (lines 96-97). Patient is underaged worldwide and cannot decide by herself legally.

Response: We would like to thank the reviewer for the comment. As the patient was underage, we had also obtained consent from the patient’s mother.

Lines 28-29: Johan et al mentioned that "20% of osteoid osteomas are located in the proximal femur" and generally in the femur as is witten in the paper.

Response: We would like to thank the reviewer for the comment. Please note that we have revised this sentence as follows:

“Approximately 20% of osteoid osteomas are located in the proximal femur [2].” (Lines 26–27)

Line 53: Please provide data regarding the general health status of the patient. Moreover in the x-rays distal femoral epiphysis does not seem normal, especially regarding the medial femoral condyle.

Response: We would like to thank the reviewer for the comment. Please note that the patient had no medical history. Regarding the activities of daily living, there were also no issues noted.

Lines 54-55: The size of the tumor as it is described is much larger than what has been already reported regarding this procedure (Fujiwara et al 2018). Is there any possibility of malpractice exceeding the limits of the procedure?

Response: We would like to thank the reviewer for the comment. We have also described the possibility of fractures after surgery as follows:

“It is essential to obtain knowledge on the material properties of the bone and biomechanics of the skeleton under a physiological load to evaluate the fracture risk related to this treatment. When the bone strength is decreased by the performed surgical procedure, the fracture may be caused by a similar compressive loading.” (Lines 113–117)

Lines 66-69: The technique has been performed in slightly different way than was described in the study of Fujiwara et al (2018) which was used as reference. In your case both cortices (lateral where tumor was located and medial) were drilled, affecting negatively the result.

Response: We would like to thank the reviewer for the comment Please refer to Figures. That is why we would like to indicate the efficacy of FEA.

Lines 70-71: Artificial bone graft was preferred over naive bone graft (allograft or autograft). Please explain which kind of artificial bone graft was used, why and the quantity of it as it cannot be located in the postoperative x-ray.

Response: We would like to thank the reviewer for the suggestions. Please note that β-tricalcium phosphate (β-TCP) block was used as a bone-grafting material. We have also corrected and improved the quality of figures.

Lines 85-86: Two cortices were drilled and almost â…“ of the femoral diameter is missing while the rest of the femoral cortex has been split in the middle. This condition caused bone weakening. Please provide data supporting the decision for not internal or external fixation immediately after the tumor resection.  Moreover, is available an x-ray at 4-week postoperatively according to it patient was advised for partial weight-bearing?

Response: We would like to thank the reviewer for the comment. To solve these issues, we would like to indicate the efficacy of FEA prior to tumor resection.

Lines 94-95: Please provide reference supporting the treatment of femoral fracture with k-wires. Even the use of titanium elastic nails is contraindicated for patients with unstable femoral fracture weighted more than 49 kg (Khoriati Al-achraf, Jones Carl, Gelfer Gael, Trompeter Alex The management of paediatric diaphysial femoral fractures: a modern approach. Strat Traum Limb Recon (2016) 11:87–97 DOI 10.1007/s11751-016-0258-2)

Response: We would like to thank the reviewer for the suggestion. The body weight of the patient was 45 kg. Moreover, the use of adjusted K-wire instead of titanium elastic nails in the intramedullary fixation of femoral shaft fractures in children is reported to be an advantageous surgical option due to the lower cost, easy accessibility and no need for a second surgery for implant removal. Thus, we have chosen the k-wire fixation. Please note that we have revised the Case Presentation section and cited the suggested study in the manuscript (Reference #16-17)

Lines 117-139: This part of discussion is irrelevant to the subject. Finite element analysis (FEA) could be useful preoperatively to prevent the fracture and the pain and the second surgery. The technology, the knowledge and the know-how was available to you as it is obvious from the self-citation

(Iwai, T.; Hoshi, M.; Oebisu, N.; Orita, K.; Shimatani, A.; Takada, N.; Nakamura, H. Prediction of risk factors for pathological fracture after bone tumor biopsy using finite element analysis. Cancer Manag Res 2021, 13, 3849–3856)

Response: We would like to thank the reviewer for the comment. We believe that FEA is an efficient method.

Round 2

Reviewer 3 Report

Dear authors,

Your effort to improve the manuscript was present but some points mentioned before were not addressed properly and adequately.

Consent form: The consent form included in the non-published material remains signed only by the underage patient. To be sincere, any reviewer could reject any manuscript with this serious fault, compromising the integrity of the journal.

Lines 32-34: In this newly added sentence use of therapeutic alternatives with natural molecules is suggested. Please explain what do you mean with natural molecules and how this reference regarding temporomandibular disorders and muscle pain is associated to the case described (osteoid osteoma).

Line 53: Please include data regarding the general health status of the patient in the manuscript. This information regards any reader not only me.

Lines 54-55: My point (“The size of the tumor as it is described is much larger than what has been already reported regarding this procedure (Fujiwara et al 2018). Is there any possibility of malpractice exceeding the limits of the procedure?”) remained unanswered. The surgical team knew the dimensions of the osteoid osteoma exceeded the higher limits for tumor resection with this technique previously described. Please explain why this case is not medical malpractice.

Lines 66-69: The surgical technique is totally different than pre- or postoperative FEA. In your case you performed a surgical technique differently than described by Fujiwara et al (2018) for a larger bone tumor, which led to a pathological fracture. The paper of Fujiwara cannot be used as reference.

Lines 78-79: Please explain why artificial bone and not autograft was chosen. Please include all important data (type of bone graft, volume, etc.) in the manuscript, not only in the response letter.

Lines 97-99: “Based on a previously report…” Your references are irrelevant to what you chose and decided. In Khoriati Al-achraf, Jones Carl, Gelfer Gael, Trompeter Alex The management of paediatric diaphysial femoral fractures: a modern approach. Strat Traum Limb Recon (2016) 11:87–97 DOI 10.1007/s11751-016-0258-2 there is no mention to K-wires as surgical method for femoral fracture in 12-year old patient. Moreover, the second reference of Isik et al (2015) presents a paidiatric population who is maximum 11 years old. Your patient is at least one year older. Please be careful when cite because copyright laws are strict.

Lines 120-165: My point was that FEA should be performed preoperatively, before the bone tumor resection and not after the pathological fracture of the femur. FEA is an efficient method, as you mentioned, to predict and prevent such catastrophic results as this young patient suffered. The optimal sequence for this 12-year-old girl could be:

  1. Clinical and radiological evaluation of the osteoid osteoma.
  2. FEA to predict bone mechanical properties after the bone resection with the decided method, modified by you according to tumor dimensions.
  3. Tumor resection and femoral stabilization with Elastic titanium Nails for instance, according to results of FEA.

In the manuscript you mentioned that “FEA requires certain level of software …and time… especially in cases of trauma”. In the case of 12-year-old patient there were no time limitations as the osteoid osteoma resection is not an urgent procedure and the proper software and know-how was available to you.

 To conclude, for second time I am not sure which is the goal of this case report. As far as I am concerned a title like“Finite Element Analysis could predict and prevent a pathological femoral shaft fracture after en bloc resection of a large osteoid osteoma” , could be more appropriate.

Author Response

Dear authors,

Your effort to improve the manuscript was present but some points mentioned before were not addressed properly and adequately.

Response: The authors would like to thank the reviewer for their constructive critique to improve the manuscript. We have made every effort to address the issues raised and to respond to all comments. The revisions are indicated in red font in the revised manuscript. Please, find next a detailed, point-by-point response to the reviewer's comments. We hope that our revisions will meet the reviewer’s expectations.

Consent form: The consent form included in the non-published material remains signed only by the underage patient. To be sincere, any reviewer could reject any manuscript with this serious fault, compromising the integrity of the journal.

Response: We would like to thank the reviewer for the comment. As the patient was underage, we had also obtained consent from the patient’s mother. Please refer to the attached document.

Lines 32-34: In this newly added sentence use of therapeutic alternatives with natural molecules is suggested. Please explain what do you mean with natural molecules and how this reference regarding temporomandibular disorders and muscle pain is associated to the case described (osteoid osteoma).

Response: We would like to thank the reviewer for the comment. As you mentioned, there is little relationship between temporomandibular and osteoid osteoma. Therefore, we have deleted the sentence and the reference in the revised manuscript.

Line 53: Please include data regarding the general health status of the patient in the manuscript. This information regards any reader not only me.

Response: We would like to thank the reviewer for the comment. As you suggested, we have added the information (Lines 53–54).

Lines 54-55: My point (“The size of the tumor as it is described is much larger than what has been already reported regarding this procedure (Fujiwara et al 2018). Is there any possibility of malpractice exceeding the limits of the procedure?”) remained unanswered. The surgical team knew the dimensions of the osteoid osteoma exceeded the higher limits for tumor resection with this technique previously described. Please explain why this case is not medical malpractice.

Response: We would like to thank the reviewer for the comment. As you mentioned, the size of the tumor in this case report was much larger than what had been already reported by Fujiwara et al regarding this procedure. Therefore, there might have been the possibility of malpractice exceeding the limits of the procedure.

Lines 66-69: The surgical technique is totally different than pre- or postoperative FEA. In your case you performed a surgical technique differently than described by Fujiwara et al (2018) for a larger bone tumor, which led to a pathological fracture. The paper of Fujiwara cannot be used as reference.

Response: We would like to thank the reviewer for the comment. As you suggested, we have deleted this paper as a reference.

Lines 78-79: Please explain why artificial bone and not autograft was chosen. Please include all important data (type of bone graft, volume, etc.) in the manuscript, not only in the response letter.

Response: We would like to thank the reviewer for the comment. We could not suggest the performance of autograft procedure considering the severity of operative invasion. Based on the previous report, we decided to perform an artificial bone graft (β-tricalcium phosphate (β-TCP) block,10 mm × 20 mm × 10 mm, superpore standard-type: Pentax New Ceramics Division, Hoya Corporation, Tokyo, Japan). Accordingly, we have added this information in the revised manuscript (Lines 78–82).

Lines 97-99: “Based on a previously report…” Your references are irrelevant to what you chose and decided. In Khoriati Al-achraf, Jones Carl, Gelfer Gael, Trompeter Alex The management of paediatric diaphysial femoral fractures: a modern approach. Strat Traum Limb Recon (2016) 11:87–97 DOI 10.1007/s11751-016-0258-2 there is no mention to K-wires as surgical method for femoral fracture in 12-year old patient. Moreover, the second reference of Isik et al (2015) presents a paidiatric population who is maximum 11 years old. Your patient is at least one year older. Please be careful when cite because copyright laws are strict.

Response: We would like to thank the reviewer for the comment. As you suggested, we have deleted the respective references. We have added two articles to the references (References #16 and 17).

Lines 120-165: My point was that FEA should be performed preoperatively, before the bone tumor resection and not after the pathological fracture of the femur. FEA is an efficient method, as you mentioned, to predict and prevent such catastrophic results as this young patient suffered. The optimal sequence for this 12-year-old girl could be:

  1. Clinical and radiological evaluation of the osteoid osteoma.
  2. FEA to predict bone mechanical properties after the bone resection with the decided method, modified by you according to tumor dimensions.
  3. Tumor resection and femoral stabilization with Elastic titanium Nails for instance, according to results of FEA.

Response: We would like to thank the reviewer for the comment. As you suggest, we have added these sentences in the Discussion section (Lines 163–172).

In the manuscript you mentioned that “FEA requires certain level of software …and time… especially in cases of trauma”. In the case of 12-year-old patient there were no time limitations as the osteoid osteoma resection is not an urgent procedure and the proper software and know-how was available to you.

Response: We would like to thank the reviewer for the comment. As you suggested, we have deleted this sentence in the revised manuscript.

 To conclude, for second time I am not sure which is the goal of this case report. As far as I am concerned a title like“Finite Element Analysis could predict and prevent a pathological femoral shaft fracture after en bloc resection of a large osteoid osteoma” , could be more appropriate.

Response: We would like to thank the reviewer for the comment. As you suggested, we have corrected the title of this case report (Lines 2-4).

Round 3

Reviewer 3 Report

Dear Authors,

All points mentioned previously were addressed adequately. The scientific soundness reached the publication standards. As far as I am concerned, only one point needs clarification.

Lines 100-102: The dimensions (width) of K-wires used should be mentioned. Moreover a sentence justifying the use of this method instead of any other available should be added. Both references cited regarding the use of K-wires highlight that this option is viable either "in lack of adequate financial resources" (Kumar et al 2014) or "provides a reasonable option for treatment in areas where material and financial resources are constrained" (Jain et al 2014). Moreover, if available, I would to see published a more recent x-ray where callus is formed or even the reduction remains. 

Author Response

Dear Authors,

Comment: All points mentioned previously were addressed adequately. The scientific soundness reached the publication standards. As far as I am concerned, only one point needs clarification.

Response: The authors would like to thank the reviewer for their constructive critique to improve the manuscript. We have made every effort to address the issues raised and to respond to all comments. The revisions are indicated in red font in the revised manuscript. Please find below detailed, point-by-point responses, to the reviewer's comments. We hope that our revisions will meet the reviewer’s expectations.

Comment: Lines 100-102: The dimensions (width) of K-wires used should be mentioned. Moreover a sentence justifying the use of this method instead of any other available should be added. Both references cited regarding the use of K-wires highlight that this option is viable either "in lack of adequate financial resources" (Kumar et al 2014) or "provides a reasonable option for treatment in areas where material and financial resources are constrained" (Jain et al 2014). Moreover, if available, I would to see published a more recent x-ray where callus is formed or even the reduction remains. 

Response: We would like to thank the reviewer for their valuable comments. As requested, we have added the dimensions of the K-wires used in the revised manuscript (Lines 100–102). 3.0 mm K-wires were inserted through the cortex of the femur. The treatment using K-wires was selected based on the patient’s constrained financial situation.

Page 4, lines 108-110:

“Based on previous reports that focused on the management of pediatric diaphysial femoral fractures [16, 17], intramedullary k-wire fixation using 3.0 mm Kirschner wires was performed for the fracture due to the patient’s constrained financial situation (Figure 5B).”

Furthermore, we replaced Figure 5B with a plain radiograph showing where the callus was formed.
